# Molecular Mechanisms of Immune Escape for Foot-and-Mouth Disease Virus

**DOI:** 10.3390/pathogens9090729

**Published:** 2020-09-04

**Authors:** Bo Yang, Xiaohui Zhang, Dajun Zhang, Jing Hou, GuoWei Xu, Chaochao Sheng, Sk Mohiuddin Choudhury, Zixiang Zhu, Dan Li, Keshan Zhang, Haixue Zheng, Xiangtao Liu

**Affiliations:** State Key Laboratory of Veterinary Etiological Biology, National Foot-and-Mouth Disease Reference Laboratory, Lanzhou Veterinary Research Institute, Chinese Academy of Agriculture Science, Lanzhou 730046, China; 2019107002@njau.edu.cn (B.Y.); zhangxh@njau.edu.cn (X.Z.); vetzdj129@163.com (D.Z.); Xiaokongm@163.com (J.H.); happyboysayxx@163.com (G.X.); shenchaochao1989@163.com (C.S.); mohiuddin.bau.vet786@gmail.com (S.M.C.); zhuzixiang@caas.cn (Z.Z.); lidan@caas.cn (D.L.); liuxiangtao@caas.cn (X.L.)

**Keywords:** foot-and-mouth disease virus, immune escape, innate immune response, interferon, protein interaction, viral protein

## Abstract

Foot-and-mouth disease virus (FMDV) causes a highly contagious vesicular disease in cloven-hoofed livestock that results in severe consequences for international trade, posing a great economic threat to agriculture. The FMDV infection antagonizes the host immune responses via different signaling pathways to achieve immune escape. Strategies to escape the cell immune system are key to effective infection and pathogenesis. This review is focused on summarizing the recent advances to understand how the proteins encoded by FMDV antagonize the host innate and adaptive immune responses.

## 1. Introduction

Foot-and-mouth disease (FMD) is an acute and highly contagious disease affecting the cloven-hoofed animals, such as pigs and cattle. The pathogen that causes FMD is known as FMD virus (FMDV), a single-stranded positive-sense RNA virus that is classified into the genus Aphthovirus in the family Picornaviridae [1,2,3]. The pathogen causes vesicular disease of mouth and feet in susceptible animals [4]. The high mutation rate of the genome of FMDV and the rapid proliferation has led to the rapid evolution of the virus and the formation of seven main serotypes [5,6,7]. The antigenic diversity among the serotypes poses challenges to the research of efficient and cross-protective vaccines [8]. The genome of FMDV contains an open reading frame (ORF) that encodes a polyprotein precursor, and it is cleaved into four structural proteins and 10 non-structural proteins by viral autoproteases and host protease [2,9] (Figure 1).

Upon infection of the host, a virus will face the attack from the host’s immune response. In the long-term battle with the host immune response, the virus has evolved and developed a series of immune escape mechanisms to overcome the killing and inhibition from the host immune system. The mechanism of virus immune escape can be divided into three categories: (1) enable the virus to avoid the recognition of humoral immune response; (2) interfere with the function of cellular immune response; (3) interfere with the host’s immune response to the virus [10]. All these strategies would be exploited by the virus for replication and spreading to other hosts.

As a highly contagious and fast-spreading virus, FMDV has multiple ways to evade the killing by the immune system [11], which makes it difficult for controlling the virus. Viral capsid protein VP1 and leading protein L^pro^ can inhibit the production of interferon (IFN) and innate immune response by interacting with soluble resistance-related calcium-binding protein (sorcin) or host transcription factor ADNP [12,13]. Recently, new mechanisms and functions of FMDV proteins inhibiting innate immunity have been discovered. DDX56 (a kind of RNA helicase), participate in RNA metabolism and ribosome synthesis is reported to involve in this new mechanism. The interaction between FMDV 3A and DDX56 suppresses the host innate immunity by reducing the phosphorylation of IRF3 [14]. In addition, nucleotide-binding oligomerization domain 2 (NOD2), a member of the nucleotide-binding oligomerization domain-like receptor (NLR) family [15], activates the NF-κB and IFN-β signaling pathways during FMDV infection and inhibits the replication of FMDV in infected cells [16]. FMDV 2B, 2C, and 3C^pro^ inhibit the expression of NOD2 protein, which antagonizes the antiviral response [16]. Reportedly, multiple structural and non-structural proteins of FMDV escape the killing of the host immune system. This review summrized the molecular mechanisms of immune evasion caused by FMDV proteins. The present study aimed to fill the gaps of knowledge on FMDV immune evasion mechanism, providing the basis for the prevention and control strategies for FMDV.

## 2. Molecular Mechanisms in FMDV Structural Protein Immune Escape

The P1 structural protein of FMDV was cleaved into three main viral structural proteins VP0, VP1 and VP3 by 3C^pro^ protease during the later translation and modification. VP0 protein was further cleaved into VP4 and VP2 proteins by protease 3C.

### 2.1. VP0

Interestingly, the VP0 protein is a cleavage precursor of VP2 and VP4 [2]. The last step in the production of mature virions is the cleavage of 1AB (VP0), which converts 85 residues of the N-terminal into VP4 and the remaining into VP2, although some copies of VP0 may be retained in the intact virions [17]. Previous studies reported that VP0 protein of FMDV inhibits the activation of type I IFN signaling pathway by interacting with IRF3 [18] (Figure 2, Table 1). However, further studies are essential to assess the combination of VP0 to IRF3 to restrain the production of IFNs. Since then, it is reported that VP0 proteins of FMDV interact with Poly (RC) binding protein 2 (PCBP2) to promote the replication of FMDV [19] (Figure 2, Table 1). The PCBP2 can recruit E3 ligase AIP4 that contains the Hect domain into the polyubiquitin and degrades MAV [20]. The VP0 protein of FMDV suppresses the host’s innate immunity by cooperating with PCBP2 to suppress the activation of IFN-β promoter. VP0 protein promotes the formation of PCBP2-virus-induced signaling adapter (VISA) complex, enhances the degradation of VISA mediated by PCBP2, and promotes the replication of FMDV [19]. In addition, the VP0 structural protein of FMDV is necessary for the correct assembly of the virus [21].

### 2.2. VP4

The FMDV capsid protein, VP0, was further cleaved into VP4 and VP2 proteins by protease 3C. According to the structure of the virus, the VP4 protein of FMDV is localized on the inner surface of the capsid [22]. Although VP4 does not stimulate the production of neutralizing antibodies independently, it contains T and B cell epitopes, which could be recognized by a variety of haplotype MHC molecules and exhibit high immunogenicity [23,24,25]. Therefore, the combination of FMDV VP4 and VP1 could be used as a backup antigen for the development of a universal vaccine [26,27]. In addition, the VP4 protein plays a crucial role in immunosuppression. The recombinant FMDV VP1–VP4 protein has been reported to have an inhibitory effect on the innate immune function of mouse peritoneal mast cells, putatively mediated by mannose receptor [28]. Furthermore, nucleoside diphosphate kinase 1 (NME1) regulates the function of p53 to prevent tumor metastasis and progression and inhibit the metastasis of several malignant tumors [29,30,31]. The role of NME1 in viral infection is not yet clarified. Recent studies have demonstrated that NME1 has antiviral activity and enhances p53-mediated transcription, while p53 regulates the expression of many antiviral genes to perform antiviral functions. However, FMDV VP4 does not directly interact but degrades NME1 through macroautophagy [32,33,34] (Figure 2, Table 1). This phenomenon promotes the interaction between p53 and MDM2 (MDM2 is a negative regulatory factor of p53), while on the other hand, it enhances the MIF-mediated inhibition of p53 activity, thereby impeding the antiviral response [34].

Autophagy is an ancient and conservative biological process, which exists in almost all eukaryotes. Through continuous research, it has been found that autophagy can selectively degrade intracellular redundant or harmful substances [35,36,37], thus affecting the pathogenesis of some diseases. Autophagy can also degrade invading microorganisms (such as bacteria, virus, and parasites) [38], and is one of the immune mechanisms against pathogenic infection. It has been proved that a variety of viruses can activate autophagy and be swallowed and degraded [39]. Not only that, after virus infection, host cells resist virus infection by releasing inflammatory factors and activating innate and adaptive immune responses, and autophagy also plays an important role in these defense responses [40]. In the long process of coexistence of virus and host, autophagy pathway has become one of the targets of virus-versus-host immunity. The inhibitory effect of virus on autophagy is also in many ways. For example, some studies have shown that after PRRSV infection, the type I microtubule-associated protein light chain 3 (LC3- I) is transformed into LC3- II, which activates the autophagy mechanism and leads to the accumulation of autophagosomes by preventing the fusion of autophagosomes and lysosomes. Autophagosomes can act as replication sites to enhance PRRSV replication [41,42].

### 2.3. VP2

VP2 is one of the structural proteins of FMDV, localized on the surface of the virus. Types O, A, and C FMDV VP2 contain several antigenic sites that present immunological significance [24,43,44,45]. In addition, the amino acid substitution on the VP2 B-C loop of FMDV type ASIA1 not only mediates the significant antigenic diversity but also alters the replication ability and pathogenicity of the virus. For example, the single Asp-to-Asn substitution at VP2 72 position will reduce the virus replication ability and virulence [46]. However, the exact reasons for this result need to be further studied. A recent study demonstrated that the interaction between FMDV VP2 and HSPB1 activates the EIF2S1-ATF4 pathway, which in turn, inhibits the AKT-MTOR pathway, leads to autophagy, and promotes virus replication [47] (Figure 2, Table 1). Autophagy has been proposed to provide a membrane platform for virus replication complexes or mediate the virus assembly and release [48]. Thus, autophagy plays a crucial role in the replication of FMDV, and the expression of related VP2 mutants decreases the level of autophagy. Therefore, VP2-induced autophagy may be one of the mechanisms of FMDV infection. Autophagy regulates type I IFN signaling machinery and plays a vital role in antiviral innate immunity [49,50], and ATG12-ATG5 conjugate inhibits the production of type I IFN during VSV infection [51]. Thus, it can be speculated that FMDV VP2 induces autophagy to increase the replication of FMDV, which might be achieved by blocking type I IFN signal. Also, the correlation between VP2-induced autophagy and host antiviral immunity of FMDV needs to be investigated further. The changes in some sites on the surface of the VP2 protein of FMDV affects the cellular tropism and adaptability of the virus; for instance, the replacement of (Glu82 to Gly) alters the binding characteristics of the virus to cells [52]. The positively charged lysine residue at the VP2131 site of FMDV A can increase the adaptability of BHK-21 cells [53]. Furthermore, the change in some sites of VP2 would also affect the stability and immunogenicity of FMDV. For example, VP2 H145Y replacement reduces the acid sensitivity of the capsid of FMDV type ASIA 1, makes the H145Y mutant of FMDV resistant to acid, and enhances the immunogenicity of virions [54]. The tyrosine at position 98 of VP2 mutated to phenylalanine (Y98F) enhanced the thermal stability of the virus. This mutant presented optimal immunogenicity, and neutralizing antibodies could be induced by immunizing guinea pigs [55]. Thus, identifying these specific sites of VP2 protein in FMDV provides an idea for the preparation of a heat-resistant and immunogenetically superior FMD antigen.

### 2.4. VP1

The VP1 protein is the major surface protein on the FMDV and the primary antigen that elicits the neutralizing antibody response. The FMDV VP1 stimulates the host to produce CD8+ T cell responses with cross-protection against multiple serotypes of FMDV. Therefore, VP1 protein or its antigenic determinants have become a research hotspot in the development of novel vaccines [2,56,57]. FMDV protein aqueous soluble recombinant DNA-derived VP1 (rVP1) binds to integrin induces apoptosis, and FMDV rVP1 may selectively act as an effective human tumor apoptosis factor by regulating Akt signal pathway [58]. The VP1 that interacts with host proteins can either enhance or inhibit the production of IFN in cells. First, it was found that VP1 interacts with soluble resistance-related calcium-binding protein (sorcin), a negative regulator in the innate immune signaling pathway, through yeast two-hybrid and immunoprecipitation experiments. Also, VP1 binds to sorcin and activates the transcription factor STAT3. STAT3 inhibits the activation of IKK and NF-κB pathway, thus inhibiting the expression of type I IFN and cytokines [12] (Figure 2, Table 1). Second, the host protein kinases are essential regulators of virus interaction and play a crucial role in virus replication. TPL2 (tumor progression locus 2), a serine/threonine-protein kinase, promotes the activation of IFN-β signaling pathway by increasing the phosphorylation of IRF3. TPL2 phosphorylation site Thr290 is vital for promoting IRF3-induced IFN-β signal activation. VP1 inhibits the protein expression of TPL2 phosphorylated at Thr290, thereby inhibiting the IRF3-activated IFN-β signaling, while VP1 reduces the mRNA levels of TPL2-mediated IFN-β and some ISGs (Figure 2, Table 1). Third, another study proved that VP1 suppresses the IFN-β signaling pathway at the IRF3 level by inhibiting the IRF3 phosphorylation, dimerization, and nuclear translocation (Figure 2, Table 1). Another study suggested that the activation of the mitogen-activated protein kinase (MAPK) pathway is essential for FMDV replication. FMDV VP1 interacts with host ribosomal protein SA (RPSA) to continually activate the MAPK signal pathway and promote virus replication by inhibiting the RPSA-mediated function [59] (Figure 2, Table 1).

### 2.5. VP3

As a structural protein of FMDV, VP3 plays a crucial role in virus assembly [9]. It blocks the IFN signal transduction, promotes the replication of FMDV, and inhibits the host immune response. In addition, VP3 inhibits the protein and mRNA expression of innate immune junction molecule, VISA. It interacts with the VISA protein to inhibit the formation of VISA-regulated complex, thereby inhibiting the dimerization and phosphorylation of IRF3, inhibiting the expression of antiviral genes induced by IFN-β, and promoting FMDV replication [60] (Figure 2, Table 1). Previous studies have focussed on the effect of FMDV after treatment of type I IFN. Also, it was demonstrated that FMDV VP3 inhibits the type II IFN signaling pathway. Furthermore, VP3 interacts with the host protein kinase JAK1 protein and degrades the JAK1 protein through lysosome pathway, inhibits the activation of the JAK-STAT pathway, and reduces the IFN-induced antiviral gene expression [61] (Figure 2, Table 1). During evolution, some host substances can act on viral proteins, inhibit viral replication, and resist infection. Reportedly, the FMDV infection stimulates the expression of miR-1307, which indirectly induces the degradation of FMDV VP3 protein through the proteasome pathway and strengthens the host immune response to inhibit the replication of FMDV [62]. Moreover, the induction of miR-1307 is earlier than the full activation of NF-κB and IRF3/7 [62]. Nonetheless, FMDV infection-stimulated expression of miR-1307 would be a research hotspot in the future. However, the direct goal of miR-1307 has not yet been determined. Recently, it has been reported that TANK-binding kinase1 (TBK1) degrades the VP3 protein of several types of small ribonucleic acid virus, including FMDV, through its kinase and E3 ubiquitin ligase activity, while p-TBK1 is highly enriched in the miR-1307 overexpression cells [63]. Thus, miR-1307 may target negative regulatory factors of TBK1. It has been reported that single or co-transfection of microRNAmiR-203a-3p and miR-203a-5p in porcine cell lines followed by infection of FMDV resulted in a decrease in viral protein synthesis and virus production. So, miR-203a-3p and miR-203a-5p are potential natural biotherapies against foot-and-mouth disease virus [64].

## 3. Molecular Mechanisms in FMDV Non-Structural Protein Immune Escape

### 3.1. L^pro^

FMDV L^pro^, a papain-like protease, is the first translated protein from the FMDV genome and coexists in two forms in vivo and in vitro, Lab^pro^ and Lb^pro^ [65,66,67]. L^pro^, a key virulence factor of FMDV, suppresses the host immune response and achieves immune escape. L^pro^ can cleave host translation-related proteins or interact with host transcription factors to inhibit the synthesis of antiviral factors. L^pro^ can target cleavage/degradation pattern recognition receptors, the proteins of the interferon pathway, NF-κB pathway, and stress-related pathway. In addition, L^pro^ acts as a deubiquitinase (DUB) and deISGylase.

First, L^pro^ specifically cleaves the eukaryotic initiation factors (eIFs), 4GI and 4GII. The loss of integrity of eIF4GI and eIF4GII hinders the formation of eukaryotic cellular translation initiation factor 4F(eIF4F) complex, while eIF4F complex significantly affects the cell cap-dependent translation [68], thereby preventing the recruitment of capped mRNA in host cells and inhibiting the synthesis of antiviral molecules of innate immunity [69]. FMDV RNA starts translation in a cap-independent manner through the internal ribosome entry site (IRES) elements [70,71,72]. Therefore, FMDV can make use of host protein synthesis mechanism to quickly synthesize virus protein and complete virus reproduction. The interaction between L^pro^ and activity-dependent neuroprotective protein (ADNP) is crucial in the process of infection and promotes FMDV replication by inhibiting the expression of IFN and IFN stimulated gene (ISG) [13]. However, whether the processing of ADNP by L^pro^ is performed directly by L^pro^ or by related enzymes induced or activated by L^pro^ is yet to be deduced. The present study further elucidates the mechanism though which FMDV evades the immune response by interaction with transcription factors.

Second, FMDV L^pro^ downregulates the expression of NF-κB and IRF3/7, which in turn, interferes with the transcription of IFN-α1/β mRNA. FMDV L^pro^ induces the degradation of p65/RelA, the core component of NF-κB, which destroys the integrity of NF-κB and downregulates the transcription of IFN-β in host cells, thus inhibiting the host immune response [73]. However, the degradation mechanism of p65/RelA by L^pro^ is still unclear, and the products of p65/RelA digested by L^pro^ have not been identified. In addition to destroying the integrity of NF-κB, L^pro^ also decreased the expression of IRF3/7, the key factors of virus-triggered IFN-α/β secretion that inhibit the production of dsRNA-induced type I IFN [74].

Third, porcine IFN-λ1, a type III IFN, inhibits the replication of FMDV. However, after screening, the lead protease L^pro^ exerts a robust inhibitory effect on the activity of IFN-λ1 promoter induced by poly(I:C) by inhibiting the RIG-I/MDA5 pathway and interfering with the activation of Interferon regulatory factors (IRFs) and NF-κB. FMDV L^pro^ alone can inhibit the expression of dsRNA-induced IFN-λ1, suggesting a new mechanism of FMDV antagonizing IFN-λ1-mediated innate immune response [75].

Fourth, in addition to its conventional papain-like protease activity, L^pro^ acts as a deubiquitinase (DUB) and deISGylase. Lb^pro^ has deubiquitination activity, and the deubiquitination functional sites are highly conserved among the seven serotypes of FMDV. These motifs could significantly inhibit the ubiquitination of key molecules of innate immune signaling pathway such as retinoic acid-inducible gene I (*RIG-I*), TANK-binding kinase 1 (*TBK1*), TNF receptor-associated factor 6 (TRAF6), and TRAF3, thereby inhibiting the innate immune response and achieving immune escape [76]. Lb^pro^ selectively cleaves the C-terminal peptide bond of ISG15 and exposes an easily detected GlyGly epitope on the substrate of the modifier, which provides a new method for monitoring FMDV [77]. A new study shows that abolishing/reducing the deISGylase/DUB activity of L^pro^ causes viral attenuation independently of its ability to block the expression of IFN and ISG mRNA [78]. The latest research shows that L^pro^’s ability to cleave RLR signaling proteins but not its deubiquitination/deISGylation activity correlates with the reduced IFN-β gene transcription [79].

Fifth, laboratory of genetics and physiology 2 (LGP2), an innate immune sensor promotes the interaction between viral RNA and MDA5, thus producing antiviral signals [80]. Recently, it has been reported that LGP2 is the biphasic main activator of many innate immune genes that induce the production of IFN by a cascade effect [81]. However, FMDV L^pro^ cleaves LGP2 and blocks the effect of LGP2-mediated IFN-β induction [82]. These features represent a new approach of immune escape and provide a basis for in-depth research on the role of LGP2 in anti-FMDV response and the interaction between MDA5 and LGP2-L^pro^.

Sixth, recent studies have demonstrated that the stress response was inhibited during FMDV infection. FMDV L^pro^ targets the SG (stress granule) scaffold proteins G3BP1 and G3BP2 to antagonize the formation of SG, a potentially significant antiviral signaling platform [83,84,85,86] that regulates the integrated stress response [87]. However, the L^pro^-mediated SG inhibition mechanism of FMDV might not be the only one in the cells infected with FMDV. Although several studies have suggested the antiviral effect of SGs, their exact role as an antiviral signal platform is not yet clarified. Therefore, these studies demonstrated that FMDV L^pro^ inhibits the host immune response and promotes the FMDV replication through various mechanisms (Figure 3, Table 2).

### 3.2. 2B

FMDV 2B protein is a hydrophobic transmembrane viroporin with oligomeric transmembrane pores that can destabilize the integrity of the host cell membrane, disrupt host cell Ca^2+^ balance, induce host cell autophagy, and promote virion release [88,89]. FMDV 2B may play an active role in virus immune escape because of its viroporin characteristics. Previous studies have shown three pattern recognition receptors, retinoic acid-inducible gene I (RIG-I), melanoma differentiation-associated factor 5 (MDA5), and LGP2 that bind to viral RNA. Of these, RIG-I and MDA5 recognize different structures of RNA to activate the antiviral signal transduction, and LGP2 regulates this process [80,90,91]. Targeted studies have found that RIG-I and LGP2 inhibit the FMDV replication, and LGP2 significantly inhibits the inflammatory response of FMDV-infected cells. FMDV 2B protein suppresses the expression of pattern recognition receptors RIG-I, MDA5, and LGP2, inhibiting the host antiviral response and promoting FMDV replication. 2B protein directly interacts, reduces the protein levels, and inhibits the antiviral effect mediated by RIG-I, MDA5, and LGP2 (Figure 3, Table 2). This reduction does not depend on proteasome, lysosome, or autophagy pathway, and the specific molecular mechanism is yet unclear. In addition, the study on whether 2B reduces the level of other junction molecules in RIG-I like receptor (RLR) signaling pathway showed that although FMDV 2B does not mediate the decline in TBK1 and IRF3 in the RLR signaling pathway, it inhibits the phosphorylation of TBK1 and IRF3, followed by suppression of the expression of type I IFN [92,93,94] (Figure 3, Table 2).

The results of yeast two-hybrid screening and immunoprecipitation showed that one of the two host proteins that could interact with FMDV 2B protein is the eukaryotic translation elongation factor1 γ (eEF1G) [95]. A previous study confirmed that the decrease in eEF1G affects the synthesis of some membrane proteins necessary for vesicle formation, and its mislocation reduces the synthesis of membrane proteins [96]. The 2B protein of FMDV is associated with increased cell membrane integrity and membrane permeability [88]. Therefore, it can be speculated that eEF1G assists 2B in producing virus-induced vesicles and inducing cell lysis. However, further studies are required to substantiate these findings. The other host protein is cyclophilin A, which degrades FMDV L^pro^ and 3A protein that suppresses the innate immune. Strikingly, the interaction between 2B protein and cyclophilin A directly inhibits the degradation of L^pro^ and 3A protein by cyclophilin A, thereby inhibiting the host immune response [97] (Figure 3, Table 2). Recent studies have shown that cyclophilin A also promotes the ubiquitination of RIG-I, and thus enhances the innate immune response [98]. However, whether the interaction between 2B protein and cyclophilin A affects the ubiquitination of RIG-I is yet to be elucidated.

Another study showed that FMDV 2B protein interacts with NOD2 to reduce the expression of NOD2 protein, for which the 2B carboxyl-terminal 105–114 region was essential, thus inhibiting the activation of NF-κB and IFN-β signaling pathways (Figure 3, Table 2). The decrease in NOD2 is not related to the cleavage of EIF4G, induction of apoptosis or proteasome, nor lysosome or caspase pathways [16].

### 3.3. 2C

FMDV protein 2C is a highly conserved polypeptide of 318 amino acids (aa) [99,100]. Subsequent studies proved that the 2C protein plays a critical role in virus replication. Guanidine hydrochloride, a molecular antagonist of protein 2C, suppresses the synthesis of viral genetic material in small ribonucleic acid virus-infected cells [101,102]. Three host proteins Beclin1, vimentin, and NOD2 interacting with FMDV 2C, were screened by yeast two-hybrid assay. Beclin1 is related to the formation of autophagosomes and the fusion of autophagosomes to lysosomes [103,104]. Protein 2C interacts and inactivates Beclin1, which in turn, inhibits the fusion of autophagosomes containing FMDV and lysosomes, thereby preventing the degradation of the virus [105] (Figure 3, Table 2). Vimentin plays a role in the lysosomal degradation of proteins and has been shown to be related to autophagosomes [106,107]. In the early stage of FMDV infection, vimentin forms a cage-like structure around 2C in order to facilitate virus replication. Additionally, the expression of the dominant-negative (DN) form of vimentin significantly reduces the replication ability of FMDV. The replication of FMDV requires a complete vimentin network. However, the exact mechanism of governing vimentin cage formation and dissolution remains to be elucidated [108]. The interaction between FMDV protein 2C and NOD2 reduces the level of NOD2 protein to help the virus evade the immune response, and the carboxyl-terminal 116–209 and 210–260 regions of 2C were essential for the interaction [16] (Figure 3, Table 2). Strikingly, the interaction between 2C and Beclin1 or NOD2 evades immune response through different pathways, which contributes to the replication of FMDV. The interaction between 2C and cellular vimentin is crucial for the replication of FMDV, albeit the specific pathway is not yet clarified.

### 3.4. 3A

The 3A protein is a conserved 153-aa polypeptide of FMDV, larger than other picornaviruses [9]. FMDV 3A protein is related to host tropism and virulence, and the deletion of 3A alters the virus tropism and virulence. Reportedly, a single amino acid change in 3A endowed FMDV with a new adaptive phenotype, and FMDV 3A protein is associated with membrane correlation and regulation of host protein secretion [9,109]. Plaque assay and virus titeration showed that the stable expression of 3A or 3AB protein enhances the replication of FMDV. However, the infection level of FMDV decreased after the transient expression of 3AB protein. These findings suggested that 3A and 3AB play a crucial role in the replication of FMDV [110].

FMDV 3A has been proved to interacts with cellular protein DCTN3 by the two-hybrid method. DCTN3 is a subunit of the dynactin complex, a cofactor for dynein. Dynactin-dynein complexes are related to the transport of intracellular organelles. The overexpression of DCTN3 and the disruption of dynactin-dynein complex significantly reduces the production of FMDV in infected cells. FMDV replication seems to require a complete dynamic protein cell pathway [111]. In the previous study, FMDV 3A protein mutantageness study based on reverse genetic technique revealed the effect of amino acid 89 mutation in 3A protein on the interaction between 3A protein and DCTN3 was detected by yeast two-hybrid technique. The data show that 3A could be bound to DCTN3 when amino acid 89 was alanine or leucine, but when amino acid 89 was mutated to proline, which destroy the binding between 3A and DCTN3 [111]. Interestingly, both the FMDV O/TAW/97 strain with PLDG peptide from 89–92 aa on 3A and the recombinant FMDV O1C3A virus with deletion of residues 87–106 on 3A lacked the binding site of the host protein DCTN3. Also, the replication rate in primary bovine cells was slower than that of parental virus strains. However, no significant change was detected in the replication rate of either of the two viruses and their parent virus strains in porcine cells [112,113]. Thus, it could be speculated that the binding of FMDV 3A and host DCTN3 might be related to the host tropism of the virus, but the slight difference in DCTN3 among species could be attributed to the range of hosts of FMDV. The recombinant FMDV with PLDG residues 89–92 on 3A produced a delayed and mild disease in cattle, suggesting that 3A-DCTN3 interaction might play a role in the virulence of the bovine virus. In addition, the virus recovered rapidly during infection and regained DCTN3 binding, suggesting that the interaction between FMDV 3A and DCTN3 is vital for virus replication in cattle [111] (Figure 3, Table 2). However, the 3A-DCTN3 combination needs further exploration.

The type I IFN reporting system was utilized to confirm that the 3A protein inhibits the activation of the IFN-β signaling pathway. Further studies demonstrated that 3A protein interacted with innate immune molecules RIG-I, MDA5, and VISA, inhibited the expression of innate immune junction molecules, such as RIG-I, MDA5, and VISA, and inhibited the formation of signal transduction complex, thereby escaping the host innate immune system [114] (Figure 3, Table 2). The overexpression of the FMDV 3A inhibited the Sendai virus-triggered activation of IRF3 [114]. A recent study found that the interaction between DEAD-box helicase 56 (DDX56) and FMDV protein 3A increases the interaction between DDX56 and IRF3 and enhances the ability of FMDV 3A to inhibit IRF3 phosphorylation (Figure 3, Table 2). Also, FMDV 3A inhibits the activation of the IFN-β promoter and ISRE by reducing the phosphorylation of IRF3 and increasing the replication of FMDV. However, the overexpression of DDX56 cannot significantly reduce the phosphorylation of IRF3. Thus, we speculated that DDX56 also inhibits the production of IFN through another different pathway, thereby promoting virus replication [14].

### 3.5. 3B

Unlike other picornaviruses that encode a single 3B copy, FMDV encodes three similar but different 3B proteins (3B1, 3B2, and 3B3), which are ubiquitous in all FMDV isolates [115]. The effective replication of FMDV in bovine cells requires a full-length 3A and three VPG (3B). The deletion of the 3B3 coding sequence adversely affects the RNA replication of FMDV, and the viral activity requires highly conserved 3B3 protein [109,116]. Although FMDV lacking 3B1 and 3B2 can also reduce the viral RNA synthesis, the growth of the virus on pig-derived cells is only slightly reduced, and the disease of pigs has also been slightly alleviated [109,117]. These studies showed that 3B3 is more critical than 3B1 and 3B2 in maintaining virus RNA replication. The efficiency of RNA replication is maximal when three 3B copies coexist, and the absence of 3B1 and 3B2 might affect the virulence and host range of FMDV. However, the integration of the three proteins in replication needs to be studied further. FMDV 3B significantly reduces the levels of IFN-β, ISG15, and IL-6 levels and the activation of IFN-β, NF-κB, and ISRE promoters induced by poly(I:C), indicating that FMDV 3B is also a viral escape protein, which inhibits the response of type I IFN in cells. Subsequently, 3B blocked the interaction between RIG-I and TRIM25, thus inhibiting the ubiquitination of RIG-1 and the formation of the RIG-I-VISA complex, which inhibits the IFN signaling pathway [118] (Figure 3, Table 2). Further studies have shown that FMDV 3B reduces the expression of type I IFN by inhibiting the VISA signaling pathway via interaction with the VISA protein (Figure 3, Table 2).

### 3.6. 3C

FMDV 3C protein has been identified as a protease. It cleaves not only most of the virus precursor proteins [9] but also the host proteins to block/inhibit the cellular defense mechanism and promote virus replication. FMDV protein 3C^pro^ cleaves the related host proteins, inhibits the transcription and translation of host cells, or promotes the translation of virus RNA, thus promoting virus replication. For example, FMDV protein 3C^pro^ is related to the cleavage of histone H3, as it removes 20 N-terminal amino acid residues from histone H3. The amino terminal of H3 is related to the regulation of chromosomal transcriptional activity in eukaryotic cells. The cleavage of 3Cpro to H3 inhibits the transcription of host cells and ultimately hinders the translation of host cells [119], which is beneficial for the virus to escape antiviral immune response (Figure 3, Table 2). In addition, FMDV protein 3C^pro^ also cleaves the host translation initiation factors, eIF4G and eIF4A, which are components of the cap-binding complex eIF4F, which inhibits the synthesis of host-related antiviral proteins [120] (Figure 3, Table 2). However, it can only partially cleave eIF4G and eIF4A but not completely block the translation of host cells [121,122]. Unlike the cleavage of eIF4G by L^pro^ in the early stage of infection, the cleavage of eIF4G and eIF4A by FMDV protein 3Cpro requires the accumulation of 3C protein. Hence, it occurs in the later stage of infection cycle, and the cutting site of eIF4G by 3C^pro^ is different from that mediated by L^pro^ [120]. In addition, the cleaving of eIF4A may be disadvantageous to the virus, and the translation of viral RNA requires eIF4A [123]. The FMDV protein 3C^pro^ can also cleave the 68-kDa Src-associated substrate during mitosis (Sam68), a unique RNA-binding protein (Figure 3, Table 2). Truncated Sam68 spreads to the cytoplasm, combines with FMDV RNA and attaches to IRES to enhance the translation of the virus RNA. However, the titer of FMDV was reduced 1000-fold after transfection of Sam68-targeted siRNA molecules, indicating Sam68 might not limit to enhancing virus translation, and Sam68 may play a variety of roles in FMDV infection [124]. FMDV protein 3C^pro^ directly or indirectly degrades vital immune molecules, thus inhibiting/blocking the expression of IFN and antiviral genes. FMDV protein 3C^pro^ can degrade pattern recognition receptors, RIG-I and LGP2, thus inhibiting the production of antiviral factors and promoting FMDV replication [92,93] (Figure 3, Table 2). It also degraded the modulator necessary for innate immune key molecule NEMO, the vital regulator of NF-κB, a junction protein necessary for activating NF-κB, and the IFN regulatory factor signaling pathway. This cleavage impaired the activation of IRFs and NF-κB and inhibited the expression of downstream antiviral genes (Figure 3, Table 2). The cysteine protease activity of 3C^pro^ is necessary for 3C^pro^ to cleave NEMO [125]. Reportedly, FMDV 3C suppresses the JAK-STAT signaling pathway, thus inhibiting the antiviral response induced by IFN. Another study found that 3C protease activity promoted the degradation of KPNA1, the nuclear localization signal receptor for tyrosine-phosphorylated STAT1, to block the nuclear translocation of STAT1/STAT2 to inhibit the JAK-STAT signaling pathway [126] (Figure 3, Table 2). Autophagy-associated protein ATG5-ATG12 is shown to be associated with the replication of FMDV. In the process of FMDV infection, ATG5-ATG12 upregulates the anti-virus NF-κB and IRF3 signaling pathways, thereby inhibiting the proliferation of FMDV. FMDV protein 3C^pro^ antagonizes the host antiviral immunity and suppresses autophagy by degrading ATG5-ATG12 [127] (Figure 3, Table 2).

Furthermore, ATG5-ATG12 enhances the expression of host protein kinase PKR, a serine-threonine kinase, which can be induced by IFN and activated by double-stranded RNA (dsRNA). Consequently, it blocks the synthesis of cell and virus protein, inhibits the replication of FMDV, and exerts a significant antiviral effect [128]. Moreover, FMDV 3C^pro^ protein induces PKR degradation through the lysosome pathway and inhibits the PKR-mediated antiviral effect by downregulating the PKR protein. No interaction occurred between FMDV 3C^pro^ and PKR [129] (Figure 3, Table 2). Recent studies have shown that 3C^pro^ decreases the level of NOD2, thus inhibiting the antiviral effect induced by NOD2, and the reduction of NOD2 induced by 3C^pro^ depends on its protease activity (Figure 3, Table 2). No interaction occurred between FMDV 3C^pro^ and NOD2 [16]. Taken together, FMDV 3C^pro^ exerts its immunosuppressive function and suppresses the innate immunity of the host via a myriad of cascades.

## 4. Molecular Mechanisms for FMDV Untranslated Region in Immune Escape

### 4.1. 5′ UTR

The 5′ Untranslated Region (5′ UTR) of FMDV is about 1300 bp, which is larger than that of several small RNA virus of the family Picornaviridae. It contains S-fragment, poly (C), pseudoknots (PKs), the cis-acting replication element (cre) and internal ribosome entry site (IRES). Studies have shown that the special structure of the 5′ untranslated region is essential for the translation and replication of the virus genome [130]. The first element of the 5′ end is the S-fragment with approximately 350 nt long, the sequence can fold and form the stem ring. it is speculated that this structure can block the function of host exonuclease, thereby maintaining the stability and replication of the virus genome [9]. S fragment is related to the interaction between virus and host protein. Some studies have confirmed the direct correlation between the degree of S fragment deletion mutation and the attenuated phenotype. The FMDV mutant with 164bp deletion in the upper part of the S fragment loop was highly attenuated in vivo [131]. FMDV with deletion of S fragment induces higher expression of IFN-β and ISG mRNA, so it is concluded that S fragment of FMDV is necessary for host cell replication and regulation of innate immune response [131]. Downstream of S-fragment is the variable length poly (C) region. Studies on the viral genome have shown that a certain threshold length of poly (C) is correlated to the viability of the virus, but there is no evidence that the length of the poly (C) chain is directly related to virulence [132]. The 3′ end of ploy (c) is pseudoknots. Some studies have shown that the deletion of 86-nt in PKS reduces the pathogenicity of O/CHA/7/2011 strain of FMDV in bovine cells and bovines, and the artificial deletion of 43 bases will reduce the pathogenicity of O/ME-SA/PanAsia strain FMDV in bovine. This deletion occurs naturally in the region of the porcinophilic Cathay topotype FMDV genome. It is suggested that the natural absence of PKs area may be the reason for the transformation from bovines to pigs as a vector for the transmission of FMDV. It is concluded that the pseudoknots region of FMDV 5′ Untranslated Region is the decisive factor of virus tendency and virulence [133]. The cis-acting replication element (cre) is a stem-loop structure containing conserved AAACA motifs [134,135]. It has been confirmed that FMDV cre is necessary for genome replication, and cre has been found to be adjacent to IRES, which indicates that cre may play a role in coordinating translation and replication, but this still needs further confirmation [136].

### 4.2. 3′ UTR

FMDV 3′-UTR consists of a 90nt structural sequence folded into two independent stem loops and a Poly (A) tail of variable length [137,138]. Related studies have shown that the 3′-UTR directly binds to the S-fragment and IRES elements of the 5′-UTR at different sites, and FMDV 3′-UTR affects virus replication and virulence by enhancing the activity of IRES elements [139]. Moreover, genetic studies have shown that the recombinant FMDV with missing structural sequence of 3′-UTR cannot be recovered [140], and it is concluded that the structural region of 3′-UTR is essential for the infectivity and replication of FMDV. In addition to the direct RNA-RNA interaction, 5′-3′-end bridging, which is also related to protein-protein and protein-RNA interactions, it has been found that cellular proteins PCBPs and p47 can directly bind to S-fragment and 3-UTR [138]. It is inferred that 5′-3′-end bridging may play an important role in the replication of FMDV.

## 5. Prospects and Future Directions

FMDV is a highly contagious virus that infects almost all cloven-hoofed animals, showing vesicles on the foot and mouth, skin erosion on the mucous membranes, fever, weight loss, pacing, and salivation, severely threatening the development of animal husbandry. However, in addition to causing acute infections and diseases, FMDV can be asymptomatic carriers in some cases, which might lead to another outbreak of FMD, making prevention and control challenging and costly. High infectivity, wide geographical distribution, wide host range, short-term immunity without serotype cross-protection, multiple modes of transmission, and persistent infection render the control and eradication of this disease rather difficult. Therefore, study the molecular mechanism underlying FMDV evading immunity is imperative for the control of an epidemic situation.

The immune system includes innate immunity and acquired immunity, which is a major protective system against the invasion of pathogenic microorganisms, surveillance, and removal of foreign bodies. FMDV suppresses the function of the immune system at the initial stage of infection, such that the virus can proliferate rapidly in the respiratory system and spread to its natural infection site [141]. In terms of evading the humoral immune system, each serotype of FMDV is prone to antigenic variation, which makes the virus escape from the neutralizing antibodies [142]. In the aspect of inhibiting cellular immune response, FMDV infection can cause the decrease of host lymphocytes and is accompanied by severe viremia, which will eventually lead to the destruction of T cells and FMDV infection inhibits the function of dendritic cells and weakens the ability of dendritic cells to process them into antigens [143,144]. Previous studies have shown that, MHC class I molecule expression on the surface of cells was suppressed at 30 min after FMDV infection, indicating that the cells infected with FMDV will immediately lose the ability to present MHC-I-related viral peptides to T lymphocytes. This mechanism would facilitate the virus escape from the host’s cytotoxic immune response. Limiting the killing effect mediated by NK cells is also an important mechanism for FMDV to evade the cellular immune response. Some studies have shown that the responsiveness of porcine NK cells decreases significantly 2–3 days after FMDV infection, and then returns to normal [145]. Strikingly, NK cells isolated from infected pigs could not secrete IFN-γ [146]. The research on FMDV interference with immune effect and suppression of innate immunity has been widely studied. Some proteins of FMDV (L^pro^,2B, 3A, 3B, 3C) can directly or indirectly act on retinoic acid-induced gene I-like receptor (RLR) to inhibit innate immunity [82,92,93,94,114]. FMDV VP0, VP3, 3A, and 3B reduce the expression of junction protein VISA at the transcriptional or protein level [19,60,114]. FMDV L^pro^, VP0, VP1, 2B, and 3A can directly or indirectly target IRF3 to inhibit interferon production [18,74,93,114]. VP3 and 3C proteins inhibit JAK-STAT signaling pathway, thus inhibiting ISGs production [61,126]. FMDV proteins L^pro^ and 3C inhibit the synthesis of antiviral molecules by cutting related factors of host transcription and translation [13,68,69,120,124]. In addition, it is interesting that L^pro^ protein can not only induce apoptosis, but also inhibit host cell apoptosis and promote virus replication, which is achieved by blocking the translation of α-IFN and inhibiting PKR synthesis [147]. FMDV protein VP2 and 2C can promote virus replication by regulating autophagy [47,105]. These mechanisms provide opportunities for rapid transcription and translation of FMDV.

In the previous studies on FMDV, HEK293 cells have been widely used in in vitro experiments because of its highly transfected efficiency. However, HEK293 cells are not FMDV susceptible cells and there are species differences between HEK293 cells and FMDV susceptible cells. Therefore, the use of HEK293 cells for FMDV-related research has some limitations.

## 6. Conclusions

In summary, FMDV has evolved a variety of ways to evade the immune response in the long-term combat with the host immune system. Although there are many breakthroughs in the research on the immune escape of FMDV, many mechanisms underlying the FMDV-affected host immunity have not yet been elucidated, and the interaction between FMDV protein and host protein need to be explored further. In addition to the interaction between the virus and host protein, exploring the mechanism of synergistic inhibition of immune response by multiple viral proteins is of great significance for the development of specific drugs and new vaccines. Previous studies mainly focused on the effect of FMDV with respect to innate immunity. However, there are a few studies on acquired immunity, and these need to be supplemented further. Also, persistent infection of FMDV needs to be investigated intensively in the future. It was the goal of the literature study to summarize the current knowledge and to point out future research directions and define the scientific questions that remain to be elucidated to gain better knowledge of immune responses against FMDV and its immune escape mechanisms.

## Figures and Tables

**Figure 1 pathogens-09-00729-f001:**
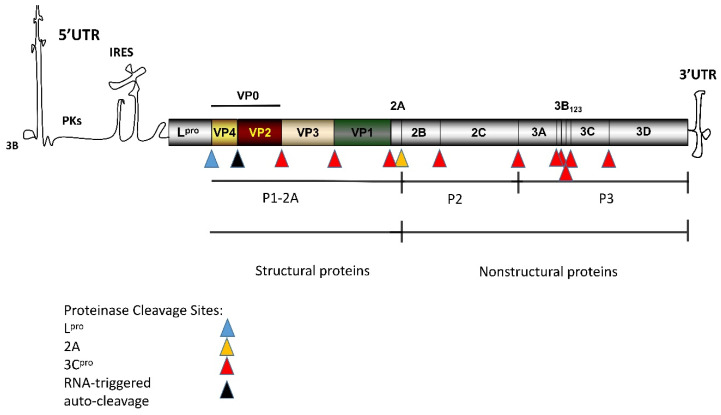
Schematic of the genome and polypeptide processing of FMDV [3,9]. The FMDV genome contains an ORF of about 7 kbp, indicated by the shaded rectangle. Each region within the ORF rectangle represents a single protein. The flank of ORF is a long 5′ untranslated region (5′-UTR) and a short 3′-UTR. 3B covalently binds to the 5′-end.

**Figure 2 pathogens-09-00729-f002:**
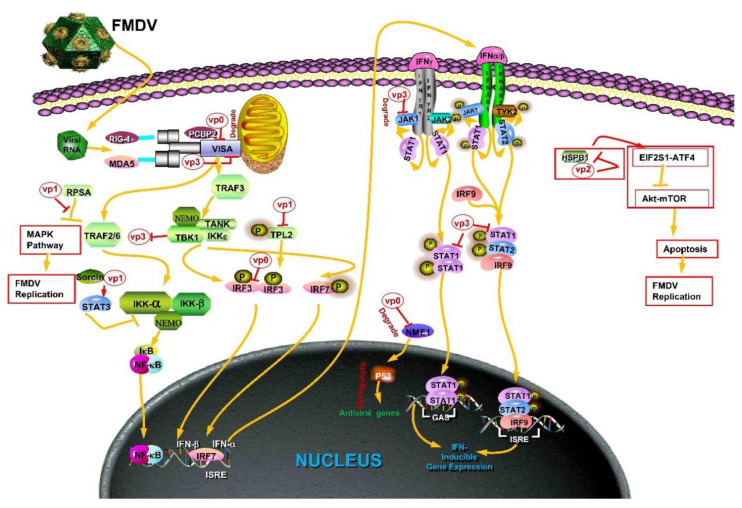
Known functions of FMDV structural proteins on immune escape. Red or yellow arrow: activation of downstream protein. Red or yellow line with vertical stub: inhibition of downstream proteins. The capsid protein of FMDV mainly targets IFN signal pathway, NF-κB pathway, JAK-STAT pathway, MAPK pathway, autophagy-related pathway and host transcription-related protein. FMDV VP0 and VP3 reduce the expression of junction protein VISA at the transcriptional or protein level. FMDV VP0 and VP1 can directly or indirectly target IRF3 to inhibit interferon production. VP3 proteins inhibit JAK-STAT signaling pathway, thus inhibiting ISGs production. FMDV protein VP2 can promote virus replication by regulating autophagy. PCBP2: poly (rC) binding protein 2; NME1: nucleoside diphosphate kinase 1; HSPB1: heat shock protein family B [small] member 1; Sorcin: soluble resistance-related calcium binding protein; TPL2: tumor progression locus 2; RPSA: ribosomal protein SA; JAK1: janus kinase 1.

**Figure 3 pathogens-09-00729-f003:**
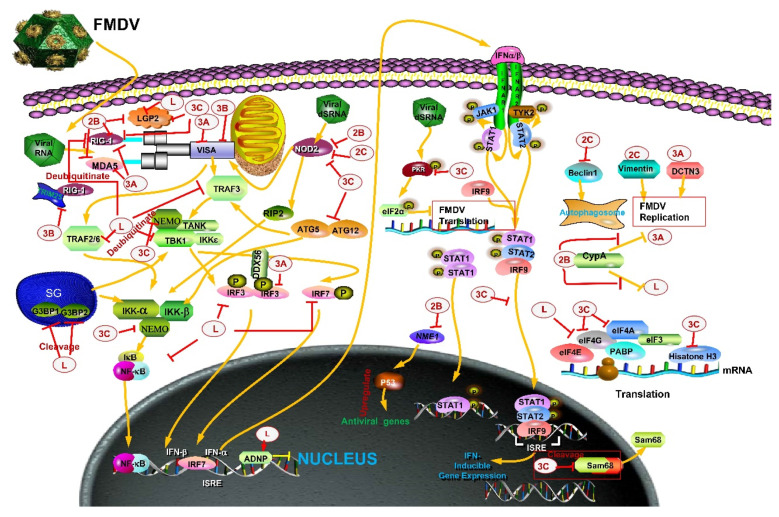
Known functions of FMDV non-structural proteins on immune escape. Red or yellow arrow: activation of downstream protein. Red or yellow line with vertical stub: inhibition of downstream proteins. As shown in the figure. FMDV structural proteins mainly targets IFN signal pathway, NF-κB pathway, JAK-STAT pathway, MAPK pathway, autophagy-related pathway and host transcription-related protein. FMDV (L^pro^, 2B, 3A, 3B, 3C) can directly or indirectly act on retinoic acid-induced gene I-like receptor (RLR) to inhibit innate immunity. FMDV 3A and 3B reduce the expression of junction protein VISA at the transcriptional or protein level. L^pro^, 2B and 3A can directly or indirectly target IRF3 to inhibit interferon production. 3C proteins inhibit JAK-STAT signaling pathway, thus inhibiting ISGs production. L^pro^ and 3C inhibit the synthesis of antiviral molecules by cutting related factors of host transcription and translation. L^pro^ protein can not only induce apoptosis, but also inhibit host cell apoptosis and promote virus replication, which is achieved by blocking the translation of α-IFN and inhibiting PKR synthesis. 2C can promote virus replication by regulating autophagy. eIF4G: eukaryotic initiation factor 4G; NF-κB: nuclear factor kappa B; IRF3: interferon regulatory factor 3; IRF7: interferon regulatory factor 7; RIG-I: retinoic acid inducible gene I; TBK1: TANK binding kinase I; TRAF6: TNF receptor-associated factor 6; TRAF3: TNF receptor-associated factor 3; ADNP: activity-dependent neuroprotective protein; LGP2: Laboratory of Genetics and Physiology 2; MDA5: melanoma differentiation associated factor 5; CypA: cyclophilin A; NOD2: nucleotide-binding oligomerization domain 2; DCTN3: dynactin 3; DDX56: DEAD-box helicase 56; Sam68: 68 kDa Src-associated substrate during mitosis; VISA: virus-induced signaling adapter.

**Table 1 pathogens-09-00729-t001:** FMDV achieves immune escape through structural protein.

FMDV Protein	Cellular Proteins	Function of Cellular Proteins	Immune Escape
VP0	IRF3	Interferon regulatory factor 3	VP0 protein inhibit the activation of type I interferon signal pathway by interacting with IRF3 [18].
	PCBP2	Poly (rC) binding protein 2	VP0 proteins interacts with PCBP2 to promote the replication of FMDV [19].
VP4	NME1	Nucleoside diphosphate kinase 1	NME1 can be degraded by VP4 through macroautophagy pathway to perform antivirus function [34].
VP2	HSPB1	Heat shock protein family B [small] member 1	Interaction between VP2 and HSPB1 activates the EIF2S1-ATF4 pathway, which leads to autophagy and promotes virus replication [47].
VP1	Sorcin	Soluble resistance-related calcium binding protein	VP1 can bind to Sorcin to inhibit the activation of IKK and NF- κ B pathway [12].
	TPL2	Tumor progression locus 2; A serine/threonine protein kinase	VP1 inhibits the protein expression of TPL2 phosphorylation site Thr290, thereby inhibiting the promotion of IRF3-activated IFN- β signal by TPL2.
	IRF3	Interferon regulatory factor 3	VP1 suppresses IFN-β signaling pathway at IRF3 level by inhibiting IRF3 phosphorylation, dimerization, and nuclear translocation.
	RPSA	Ribosomal protein SA	VP1 interacts with RPSA to maintain the activation of MAPK signal pathway and promote virus replication [59].
VP3	VISA	Innate immune junction molecule	VP3 inhibits the expression of VISA protein mRNA, and interacts with VISA protein to inhibit the formation of VISA-regulated complex, thereby inhibiting the dimerization and phosphorylation of IRF3 [60].
	JAK1	Janus kinase 1	VP3 can interact with JAK1 protein and degrade JAK1 protein to inhibit the activation of JAK-STAT pathway [61].

**Table 2 pathogens-09-00729-t002:** FMDV achieves immune escape through non-structural protein.

FMDV Protein	Cellular Proteins	Function of Cellular Proteins	Immune Escape
L^pro^	eIF4G	eukaryotic initiation factor 4G	L^pro^ cut eIF4GI and eIF4GII, thus preventing the recruitment of capped mRNA and inhibiting the synthesis of antiviral molecules [68,69].
	NF-κB	Nuclear factor kappa B	L^pro^ induce the degradation of p65/RelA, which is the core component of NF-κB [73].
	IRF3/7	Interferon regulatory factor 3/7	L^pro^ decreased the expression of IRF3/7 to inhibit the production of type I IFN induced by dsRNA [74].
	RIG-I TBK1 TRAF6 TRAF3	Retinoic acid inducible gene I; TANK binding kinase I; TNF receptor associated factor 6 TNF receptor associated factor 3	L^pro^ can significantly inhibit the ubiquitination of key molecules of innate immune signaling pathway such as RIG-I, TBK1, TRAF6, and TRAF3 [76].
	ADNP	Activity dependent neuroprotective protein	L^pro^ and ADNP interact to promote the replication of FMDV by inhibiting the expression of IFN and ISG [13].
	LGP2	Laboratory of Genetics and Physiology2	L^pro^ can cleave LGP2 and block the effect of LGP2-mediated the production of IFN-β [82].
	G3BP1 and G3BP2	stress granule scaffold proteins	L^pro^ targets to cleave the SG scaffold proteins G3BP1 and G3BP2 to antagonize the formation of SG [87].
2B	RIG-I, MDA5 and LGP2	Retinoic acid inducible gene I; melanoma differentiation associated factor 5; Laboratory of Genetics and Physiology 2;	2B protein suppress the expression of RIG-I, MDA5 and LGP2, inhibiting host antiviral response [93,94].
	TBK1; IRF3	TANK binding kinase I; Interferon regulatory factor 3	2B suppress the phosphorylation of TBK1 and IRF3, and then inhibit the expression of type I interferon [93].
	CypA	cyclophilin A	the interaction between 2B protein and cyclophilin A directly inhibits the degradation of L^pro^ and 3A protein by cyclophilin A [97].
	NOD2	nucleotide-binding oligomerization domain 2	2B protein can interact with NOD2 to reduce the protein level of NOD2, which inhibit the activation of NF- κB and IFN- β signal pathways [16].
2C	Beclin1	involve in the fusion of autophagosomes to lysosomes	2C interacts with Beclin1 to induce Beclin1 inactivation, which inhibits the fusion of autophagosomes of containing FMDV and lysosomes [105].
	NOD2	nucleotide-binding oligomerization domain 2	The interaction between FMDV protein 2C and NOD2 reduces NOD2 at the protein level to help the virus evade immune response [16].
3A	DCTN3	dynactin 3; a subunit of the dynactin complex	3A-DCTN3 interaction may play a part in the virulence of bovine virus [111].
	RIG-I, MDA5 and VISA	innate immune molecules	3A protein interacted with RIG-I, MDA5 and VISA, to inhibited the expression of RIG-I, MDA5 and VISA protein mRNA [114].
	DDX56	DEAD-Box Helicase 56	3A protein increases the interaction between DDX56 to inhibits the activation of IFN- β promoter and ISRE by reducing the phosphorylation of IRF3 [14].
3B	RIG-I and TRIM25	Retinoic acid inducible gene I;	3B blocked the interaction between RIG-I and TRIM25, thus inhibiting the interferon signal pathway [118].
	VISA	Virus-induced signaling adapter	3B reduces the expression of type I interferon by inhibiting VISA signal pathway by interacting with VISA protein.
3C	Histone H3	Related to the transcription of host cells	The cleavage of 3C^pro^ to H3 inhibits the transcription of host cells and ultimately hinders the translation of host cells [119].
	eIF4G and eIF4A	Host translation initiation factors	3C^pro^ is involved in the cleavage of eIF4G and eIF4A, thus inhibiting the synthesis of host-related antiviral proteins [120].
	Sam68	68 kDa Src-associated substrate during mitosis	3C^pro^ can also cleave Sam68. Truncated Sam68 spreads to the cytoplasm and meets the FMDV RNA and attaches to the IRES to enhance the translation of the virus RNA [124].
	RIG-I and LGP2	Retinoic acid inducible gene I; Laboratory of Genetics and Physiology 2;	protein 3C^pro^ can degrade RIG-I and LGP2 [92,93].
	NEMO	NF-κB necessary regulator	3C can also degrade NEMO to impaired activation of IRFs and NF-κB [125].
	KPNA1	the nuclear localization signal receptor for tyrosine-phosphorylated STAT1	3C promoted the degradation of KPNA1 to block the nuclear translocation of STAT1/STAT2 to inhibit JAK-STAT signal pathway [126].
	ATG5-ATG12	Autophagy associated protein	FMDV protein 3C^pro^ antagonizes host antiviral immunity and suppresses autophagy by degrading ATG5-ATG12 [127].
	PKR	a serine-threonine kinase	3C^pro^ induces PKR degradation and inhibits PKR-mediated antiviral effect by down-regulating PKR protein [129].
	NOD2	nucleotide-binding oligomerization domain 2	3C^pro^ induces the reduction of NOD2, thus inhibiting the antiviral effect induced by NOD2 [16].

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
