# Peer review of "Molecular Mechanisms of Immune Escape for Foot-and-Mouth Disease Virus"

_pathogens, 2020, doi:10.3390/pathogens9090729_

Round 1

Reviewer 1 Report

This review summarizes recent knowledge on immune escape mechanisms of FMDV. This serious disease of kloven-hoofed ungulates is a severe threat for food animal production. The literature study of Yang et al. is a very helpful summary of the very complex organisation and biology of FMDV. Also the authors made big efforts to illustrate the complex mechanisms. This is nice to have, but the construction of the table is more helpful for understanding this complex mechnisms.

Please find my comments directly in the PDF document attached.

Nevertheless, great work!

Author Response

请参阅附件。

Reviewer 2 Report

This manuscript is quite important to summarize the recent studies regarding the mechanism of immune escape for foot-and-mouth disease virus. The content is scientifically sound.  

I have no other comments.

Reviewer 3 Report

The review article by Yang et al provides a nice review on the different countermeasure mechanisms of FMDV during infection. Although the manuscript is mainly structured according to the FMDV-encoded proteins (e.g. Lpro, 3Cpro, 2C etc.), this results in redundant and overlapping pathways that are targeted by the different viral proteins. The authors should restructure the organization of this review and provide a general introduction on the different immunological pathways followed by the different FMDV evasion strategies for the particular signaling pathway discussed. This type of organization will be useful for readers that do not necessarily work with FMDV but other RNA viruses.

Some of the findings discussed in this review related to the different proteins or signaling pathways targeted by specific viral proteins have been conducted in HEK293 cells by performing overexpression of viral proteins. The authors should discuss the pitfalls of conducting this type of experiments when FMDV susceptible cells are not utilized in in vitro assays.

Other points:

Lines 16 and 19: the way the abstract is written suggest that the article is an original research. The authors need to revise the abstract.

Line 42: please revise sentence.

Line 43: it is not clear what is collectively known as “immune escape”

Line 44: unclear what is inhibiting the host immune system. Revise sentence.

Line 46: immune effect is too broad. The authors need to define “immune effect” in the context of viral infections.

Line 52: the authors need to provide some context for the statement: “The inhibition of innate immunity by FMDV should be considered during the virus replication in the early stage of infection”

Line 54: different mechanisms that induced immunosuppression by FMDV should be more specific. What type of immunosuppression: innate immunity, humoral or cellular.

Line 55-56: DDX56 and NOD2 functions should be described.

Line 61: change “results” for “data”.

Line 65: revise sentence.

Line 73: what is upstream or downstream of innate immunity? These aspects of innate immunity have not been addressed in this manuscript.

There are seven points describing the different functions of Lpro. A better introduction for these seven points should be added for clarity.  

 Line 75: “evade innate immune response through the mechanism of host protein synthesis”. This sentence should be revised as it is confusing.

Line 80-82: “restriction site” usually refers to sites in DNA molecules target by restriction enzymes. FMDV Lpro is not a restriction enzyme.

Line 85: porcine IFN-λ is not newly discovered

Line 88: typo NF-KB

Line 91: the authors should also mention the work by Swatek et al , Medina et al and Visser et al on the function of FMDV Lpro as a deISGylase.

Line 115: replate “these results” for “these studies”

Figure 2: a brief description of the figure should add clarity of Lpro functions in targeting innate immunity signaling pathways.

Line 137: Revise sentence: The PCBP2 which belongs to both RNA and DNA, …

Line 142: last sentence should be part of the introduction when describing VP0

Line 148: provide reference for: “it has some T and B cell epitopes, which could be recognized by a variety of haplotype 148 MHC molecules and has immunogenicity”

Line 159: the authors should add a few references in describing authophagy as a cellular process that is involved in regulating certain innate immune/antiviral functions.

Line 202: revise sentence. It is very unclear

Line 235: It is worth mentioning that Gutkoska et al 2017 had also reported the role of miRNA in the inhibition of FMDV replication.

Figures 2 and 4 are redundant since Lpro is a non-structural protein 

The role of untranslated regions of the genome was not discussed as a mechanism in regulating immune responses. 

Round 2

Reviewer 3 Report

The authors have addressed most of the issues raised during the first round of review. However, the manuscript still lacks clarity and organization. I highly encourage the authors to ensure that prior to submission, a native English speaker revises and edits the text. There are several confusing sentence constructions and sections that do not truly adhere with the focus of this review. Legend for Figures are missing and it is hard to follow the text. 

Please see my specific comments below:

Line 5: change; author’s name is capitalized

Line 15-21: consider revising to: Strategies to escape the cell immune system are key to effective infection and pathogenesis. This review is focused in summarizing the recent advances to understand how the proteins encoded by FMDV antagonize the host innate and adaptive immune responses.

Figure 1: typo IRSE-change to IRES. There are two FMDV genomic organization structures illustrated. This is redundant.

Line 61-63: nothing is mentioned about the function of NOD2. The authors need to indicate why does the viral protein from FMDV 3A interacts with NOD2 and how it relates to immunological escape or immunological regulation

Line 62: avoid using year dates to offer a chronological report on virus-host protein interactions discoveries. If this is something that the authors would like to highlight, a table summarizing these discoveries by date seems more appropriate.

Line 140: “According to the sequence of virus genomes, FMDV structural proteins were introduced one by one”. This sentence needs to be revised. It is hard for me to understand what the authors are trying to say.

Line 143: the authors should consider addition VP0 as part of Figure 1.

Line 177-179: “Through continuous research, it has been found that autophagy also has the function of selective degradation[[36,37], which can selectively degrade intracellular redundant or harmful substances[38], thus affecting the pathogenesis of some diseases” change to : “Through continuous research, it has been found that autophagy can selectively degrade intracellular redundant or harmful substances[36-38], thus affecting the pathogenesis of some diseases”

Line 185-191: revised paragraph. It is very confusing the example provided.

Line 266: of “small ribonucleic acid virus VP3 proteins”-revise sentence. VP3 are structural proteins and not a small ribonucleic acid protein.

Line 330-333: this sentence needs to be paraphrased this sentence. Most of the sentence is taken verbatim from Swatek et al 2018.

Line 565:  revise “cregion”

Line 573-577: very unclear whether the authors are addressing the different properties of the IRES as a factor in immunological evasion.

Line 660: change “known” to “current”

Author Response

Please see the attachment, thank you. I wish you all the best.
